# Blood Eosinophils Are Associated with Efficacy of Targeted Therapy in Patients with Advanced Melanoma

**DOI:** 10.3390/cancers14092294

**Published:** 2022-05-04

**Authors:** Simone Wendlinger, Jonas Wohlfarth, Sophia Kreft, Claudia Siedel, Teresa Kilian, Ulrich Dischinger, Markus V. Heppt, Kilian Wistuba-Hamprecht, Friedegund Meier, Matthias Goebeler, Dirk Schadendorf, Anja Gesierich, Corinna Kosnopfel, Bastian Schilling

**Affiliations:** 1Department of Dermatology, University Hospital Würzburg, 97080 Würzburg, Germany; wohlfarth_j@ukw.de (J.W.); sophia.kreft@uk-essen.de (S.K.); siedel_c@ukw.de (C.S.); teresa.kilian@stud-mail.uni-wuerzburg.de (T.K.); goebeler_m1@ukw.de (M.G.); gesierich_a@ukw.de (A.G.); corinnaveronica.kosnopfel@ukmuenster.de (C.K.); 2Department of Dermatology, University Hospital, University of Duisburg-Essen and German Cancer Consortium, Partner Site Essen, 45147 Essen, Germany; dirk.schadendorf@uk-essen.de; 3Department of Endocrinology and Diabetology, University Hospital Würzburg, 97080 Würzburg, Germany; dischinger_u@ukw.de; 4Department of Dermatology, Universitätsklinikum Erlangen, Friedrich-Alexander-Universität Erlangen-Nürnberg, 91054 Erlangen, Germany; markus.heppt@uk-erlangen.de; 5Department of Dermatology, University Hospital Tübingen, 72076 Tübingen, Germany; kilian.wistuba-hamprecht@uni-tuebingen.de; 6Department of Immunology, University Hospital Tübingen, 72076 Tübingen, Germany; 7Section for Clinical Bioinformatics, Department for Internal Medicine I, University Tübingen, 72076 Tübingen, Germany; 8Department of Dermatology, University Hospital Carl Gustav Carus at the Technical University Dresden, 01307 Dresden, Germany; friedegund.meier@uniklinikum-dresden.de

**Keywords:** melanoma, eosinophils, biomarker

## Abstract

**Simple Summary:**

Despite the advances in treatment of patients diagnosed with advanced melanoma, long-term benefits remain limited due to primary and acquired resistance. Reliable biomarkers may support treatment decisions and should optimize treatment efficacy. By using a homogeneous population of melanoma patients, peripheral blood eosinophils, along with their cytotoxic potential and soluble markers, were evaluated for their suitability as biomarkers in patients receiving targeted therapy. High relative eosinophil (REC) counts correlated with the response to targeted therapy. In vitro experiments underlined these results showing high cytotoxicity of eosinophils towards melanoma cells, which was significantly enhanced by the addition of using targeted therapy agents. We also provide evidence of a bidirectional relationship between eosinophils and melanoma cells, which might further improve the treatment of advanced melanoma.

**Abstract:**

Background: Eosinophils appear to contribute to the efficacy of immunotherapy and their frequency was suggested as a predictive biomarker. Whether this observation could be transferred to patients treated with targeted therapy remains unknown. Methods: Blood and serum samples of healthy controls and 216 patients with advanced melanoma were prospectively and retrospectively collected. Freshly isolated eosinophils were phenotypically characterized by flow cytometry and co-cultured in vitro with melanoma cells to assess cytotoxicity. Soluble serum markers and peripheral blood counts were used for correlative studies. Results: Eosinophil-mediated cytotoxicity towards melanoma cells, as well as phenotypic characteristics, were similar when comparing healthy donors and patients. However, high relative pre-treatment eosinophil counts were significantly associated with response to MAPKi (*p* = 0.013). Eosinophil-mediated cytotoxicity towards melanoma cells is dose-dependent and requires proximity of eosinophils and their target in vitro. Treatment with targeted therapy in the presence of eosinophils results in an additive tumoricidal effect. Additionally, melanoma cells affected eosinophil phenotype upon co-culture. Conclusion: High pre-treatment eosinophil counts in advanced melanoma patients were associated with a significantly improved response to MAPKi. Functionally, eosinophils show potent cytotoxicity towards melanoma cells, which can be reinforced by MAPKi. Further studies are needed to unravel the molecular mechanisms of our observations.

## 1. Introduction

Eosinophils are pleiotropic effector cells of the innate immune system. First described as operators in helminth infections and in allergic diseases, eosinophils are nowadays appreciated as effector cells with numerous immunoregulatory and inflammatory roles in asthma, allergy, cancer and even obesity [1,2,3,4,5,6]. They are equipped with specific secretory granules containing high amounts of various pro-inflammatory and lytic proteins. Major basic protein (MBP), eosinophil peroxidase (EPO), eosinophil-derived neurotoxin (EDN) [7] and eosinophil cationic protein (ECP) are released by activated eosinophils and exert their toxic capacities against healthy and cancerous tissues [8,9]. Transmembrane pores in cells are especially induced by ECP, an RNase-3 from the RNase A superfamily [10,11], causing an influx of further cytotoxic molecules leading to cell death of the target cells [12]. While we are not fully understanding the underlying mechanism, anti-tumoral cytotoxicity may, at least partially, be attributed to degranulation of eosinophils [13,14]. Whether eosinophil-derived soluble factors can predict outcomes of cancer patients is largely unknown. Despite its cytotoxic value and anti-tumor potential, high serum levels of ECP correlated counterintuitively with a worse outcome in patients with metastatic melanoma [15]. A previous study suggested absolute ECP serum levels as a novel prognostic marker by analyzing a heterogeneous patient cohort [16]. Aside from their granules, eosinophils also produce, store and rapidly secrete various immunoregulatory cytokines upon activation, for instance IFN-γ and TNF, that harbor tumoricidal effects [17,18,19,20,21]. A mouse model revealed the importance of eosinophils in improving the infiltration of CD8^+^ T cells into the tumor by CCL5, CXCL9 and CXCL10 production, thus referring to their active participation in tumor rejection [5]. Despite the early observation of tumor-associated eosinophilia in 1893 [22], later confirmed in gastric carcinoma by electron microscopy [13,23,24], it was only recently that researchers identified the interaction mechanism that drives the conjugation of eosinophils and cancer cells. In 2010, Legrand et al. described a CD11a/CD18-dependency of eosinophil-mediated cytotoxicity towards the human colon carcinoma cell line Colo-205 [18,25,26]. Just recently, activation of eosinophils via IL-33 stimulation was shown to result in stable aggregation with cancer cells in a melanoma mouse model. The study revealed the potential of immunological synapses driven by adhesion via CD11b and CD18 and the release of eosinophil-cationic protein (ECP) and granzyme-B as effective mediators with the ultimate elimination of target cancer cells [27]. How to optimally utilize the knowledge of eosinophil function and their anti-tumor capacity for the management of cancer patients in daily practice has yet to be unraveled.

There is an ongoing controversy regarding the prognostic value of eosinophilia observed in cancer patients. Eosinophil accumulation in cancer patients has been frequently described [28]. While tissue eosinophils have been linked to a poor prognosis in Hodgkin’s lymphoma [29] and various other cancers [30,31,32], patients suffering from prostate cancer [33,34] or colon carcinoma [35,36,37] benefit from high blood eosinophil counts. In melanoma, the eosinophil count has been shown to positively correlate with survival and better response to immune checkpoint inhibition (ICI) or IL-2 [38,39,40,41,42,43,44,45,46]. An early increase in eosinophil counts during anti-PD-1/anti-PD-L1 treatment was associated with an improved outcome [40,47]. Additionally, Simon et al. reported an eosinophil accumulation in the peripheral blood in responders receiving immunotherapy treatment [48]. Thus, the value of peripheral eosinophilia as a prognostic biomarker seems to be dependent on the underlying disease. Whether peripheral eosinophil levels are predictive in therapies other than ICI or IL-2 in melanoma is yet unknown.

In this study, we aim to provide insights into the bidirectional functional relationship between blood eosinophils and melanoma cells under an unattached, bloodstream mimicking condition, to fathom the clinical benefit attributed to eosinophils. We phenotypically and functionally analyzed the cytotoxic activity of eosinophils from melanoma patients and healthy donors towards a melanoma cell line model. Additionally, we specifically evaluated ECP serum levels before and upon treatment with MAP-kinase inhibitors (MAPKi). Routinely obtained laboratory values were used to determine a possible correlation with the clinical presentation.

## 2. Materials and Methods

### 2.1. Patient Cohort and Healthy Donors

All consecutive patients with newly diagnosed metastatic or unresectable cutaneous melanoma presenting at the Department of Dermatology, University Hospital Würzburg, were enrolled in this study (Table 1). Enrollment was not restricted to certain lines of therapy. Patients undergoing adjuvant therapy were not included. Patients with metastatic melanoma and no evidence of disease at the time of blood being drawn were identified in the database and enrolled during follow-up visits. In addition, all consecutive patients with newly diagnosed stage I or II melanoma were enrolled after surgical treatment. None of the early-stage patients received therapy. Patients were enrolled between July 2015 and June 2021. The study had been approved by the ethics committee of the University of Würzburg (50/17-mk). Additionally, we received 21 retrospective serum samples from advanced melanoma patients receiving targeted therapy as a first-line therapy from the Department of Dermatology, University Hospital Erlangen, and nine serum samples from the multi-centric blood bank of the Department of Dermatology, University Hospital Tübingen that were derived from patients recruited at the Department of Dermatology, University Hospital Dresden. All patients enrolled provided written informed consent. Serum samples were collected prior to and during treatment. For samples from patients receiving dual targeted therapy, the median of days between the first sample (pre-treatment) and second (on-treatment) sample was 98 days (range 58–178 days). On-treatment serum samples were obtained close to the first response assessment. For samples from patients receiving immunotherapy, the median time span was 175 days (range 52–269 days). To obtain ECP reference values, three healthy volunteers were included. White blood cell count and serum lactate dehydrogenase (LDH) were assessed 0–63 days prior to collection of the pre-treatment ECP samples and 0–63 days prior to collection of the on-treatment serum ECP samples. The closest peripheral blood draw was considered when multiple values were available. Responders were defined by RECIST 1.1. as CR (complete remission) or PR (partial remission) and non-responders as PD (progressive disease) or SD (stable disease) to the respective treatment. Demographic and clinical data were collected from all patients listed in Table 1. For phenotypical analysis of eosinophils and assessment of cytotoxicity, a second cohort of 13 samples from patients diagnosed with stage IV melanoma before the implementation of therapy, six patients with no evidence of disease (NED) in stage I and II and a total of 12 healthy donors (HD) serving as controls were included. Clinical parameters were not collected for the second cohort (Table 2). 

### 2.2. Isolation and Purification of Eosinophils

Eosinophils obtained from healthy volunteers and patients were isolated from heparinized peripheral blood samples collected in sterile tubes. Polymorphonuclear leukocytes (PMNs) were separated by density-gradient centrifugation. Blood samples diluted 1:1 with PBS were layered on Biocoll (density 1.077 g/mL, Biochrom, Berlin, Germany) and centrifuged at 360 g for 20 min at room temperature. To separate PMNs from erythrocytes, the PMN/erythrocyte suspension was incubated with a hypotonic RBC lysis buffer (Biolegend, San Diego, CA, USA) for 10 min including washing steps. Eosinophils were purified using an automatic magnetic labelling-based system, autoMACS pro, with a multi-antibody eosinophil isolation kit from Miltenyi Biotec (Miltenyi Biotec, Bergisch Gladbach, Germany). Eosinophils were purified by depletion of non-eosinophils. According to the manufacturer, the depletion cocktail contains antibodies against CD2, CD14, CD16, CD19, CD56, CD123 and CD235a (Glycophorin A).

### 2.3. Purity and Phenotyping of Eosinophils

The purity of isolated eosinophils and their phenotypic characterization was evaluated by flow cytometry using the Canto^TM^ II FACS device from BD. Dry eosinophil pellets were stained using the following antibodies to ascertain purity after isolation and separation: anti-CD16-FITC (Biolegend), anti-CD66b-APC (eBioscience, Carlsbad, CA, USA), anti-CD14-PerCP-Cy5.5 (Biolegend), anti-CD193-APC-Cy7 (CCR3, Biolegend), anti-CD45-PE-Cy7 (Biolegend), anti-CD56-PE (eBioscience), anti-CD3-PE (Biolegend) and anti-CD19-PE (Biolegend). Eosinophils were identified as being CD45+/CD16-/CD66b+/CD193+. A high purity of ≥90% was routinely obtained. For phenotypic surface staining, 5 × 10^5^ eosinophils were stained with the following antibodies to define the eosinophil population: anti-CD16-FITC (Biolegend) or –PB (Biolegend), anti-CD66b-APC (eBioscience) and anti-CD193-PE (Biolegend) or -APC-Cy7 (CCR3, Biolegend). These lineage antibodies were combined with two or three fluorochrome-labeled antibodies for the following target epitopes: HLA-DR (Biolegend), HLA-A/B/C (Biolegend), PD-L1 (Biolegend), Siglec-8 (Biolegend), TNFR2 (Biolegend), CD49d (Biolegend), CD69 (Biolegend), CD66b (eBioscience), CD31 (Biolegend) and CD29 (Biolegend). To reduce intraday variability, eosinophils from healthy donors and patients were measured the same day whenever feasible. To evaluate the effect of the metastatic melanoma cell line MaMel63a on the eosinophil phenotype, after 24 and 48 h of in vitro co-culture with a target to effector cell ratio of 1:7.5, cultured cells were washed once and stained as described. MaMel63a cells were labelled with carboxyfluorescein succinimidyl ester (CFSE, ThermoFisher Scientific, Eugene, OR, USA) prior co-culture to distinguish them from eosinophils in the co-culture. Measurements were performed using the Canto^TM^ II FACS device from BD.

### 2.4. Cell Lines

Cell lines, including the primary human melanoma cell line MaMel63a carrying the BRAFV600E mutation, were derived from patient biopsies as described previously [49]. To evaluate the sensitivity of different cancer cells to eosinophils, the non-small-cell lung cancer cell line H460 and the Merkel cell carcinoma cell line WaGa, kindly provided by D. Schrama and R. Houben (University Hospital Würzburg), were used in this study, as non-melanoma-derived and non-BRAF-mutated cell lines. Further BRAF-mutated melanoma cell lines (MaMel51, MaMel06 and MaMel80a) served as controls. Cells were grown at 37 °C with 5% CO_2_ in RPMI-1640 medium (Sigma-Aldrich, St. Louis, MO, USA) containing 10% FCS (Sigma-Aldrich) and 1% penicillin/streptomycin (Sigma-Aldrich, St. Louis, MO, USA), referred to as the complete medium (CM). All cell lines were routinely tested for the presence of mycoplasma infection and found to be negative.

### 2.5. Cytotoxicity Assays

MaMel63a cells were labelled with CFSE (2 µM, ThermoFisher Scientific, Eugene, OR, USA). Freshly isolated eosinophils were co-cultured with 2 × 10^4^ CFSE-labelled MaMel63a cells at a target to eosinophil (T:E) ratio of 1:1, 1:5, 1:7.5 and 1:10. To assess the cytotoxicity of patient-derived eosinophils pre- and on-treatment, eosinophils were isolated before the implementation of therapy and 6, 12, 24, and 48 weeks after the first assessment, and co-cultured with CFSE-stained MaMel63a cells as indicated. Eosinophils from three healthy donors served as a control cohort. Co-cultures were maintained under non-adherent culture conditions in polypropylene tubes (Beckman Coulter, Woerden, The Netherlands) or under adherent conditions in 24-well flat-bottom plates (Greiner Bio-One, Frickenhausen, Germany) as mixed culture, for 24 or 48 h as indicated, in CM or in CM containing 1 µM vemurafenib (PLX4032, Cayman Chemical, Ann Arbor, MI, USA) and/or 0.1 µM cobimetinib (GDC-0973, Cayman Chemical). For transwell experiments, CFSE-stained MaMel63a cells were cultured separately or together with freshly isolated eosinophils in polypropylene Spin-X^®^ columns equipped with an insert containing a 0.22 µm semipermeable membrane (Corning Costar^®^, Cambridge, MA, USA). Subsequently, the viability of MaMel63a cells or eosinophils was determined by 7-Amino-Actinomycin (7-AAD, ThermoFisher Scientific) and Annexin V-APC (BD Biosciences, East Rutherford, NJ, USA) staining for 15 min at room temperature. Necrosis was defined by 7-AAD-positive cells, late apoptosis was defined by the percentage of 7-AAD- and Annexin V-positive cells, and early apoptosis by Annexin V positive cells. Viable cells were defined as being 7-AAD-/Annexin V-double negative. Measurements were performed using the Canto^TM^ II flow cytometry system from BD.

### 2.6. Lysis of Eosinophils and Inactivation of Contents

To expose the intracellular eosinophil granule content, eosinophils were frozen in liquid nitrogen for 1 min as dry pellets (similar approach as published by Mattes et al. [50]). Samples were thawed at 37 °C for 5 min. Neutralization of eosinophil content after lysis was carried out by denaturation at 95 °C for 1 h utilizing a block heater (Stuart). Lysed or heat-inactivated eosinophils were resuspended in CM and cultured with CFSE-stained melanoma cells for the indicated duration.

### 2.7. Cytospins and HE staining

Non-adherent 24 h and 48 h melanoma cell–eosinophil co-cultures were performed as described above. Next, cells were transferred into the cytospin adapter (Cytospin 2, Shandon) including a glass slide (SuperFrost^®^ Plus microscope slides, R. Langenbrinck GmbH, Emmendingen, Germany), onto which cells were applied during centrifugation at 19 g for 6 min with low acceleration. Cells were dried for 15 min at room temperature and stored at 4 °C until further use. For visualization, cells were stained with hematoxylin and eosin (HE) according to the Autostainer XL protocol provided by the Dermatohistopathology Department of the University Hospital Würzburg. Imaging was carried out using a Nikon TI-E microscope.

### 2.8. Proliferation and Cell Cycle Assays

To investigate the effect of eosinophils on melanoma cell proliferation and the cell cycle under non-adherent culture conditions, MaMel63a cells were stained with Ki-67-specific antibodies or propidium iodide (PI) after 24 h of (co-)culture with eosinophils. Cells were washed with PBS after culture. For Ki-67 staining, the dry pellets were resuspended in cold 70% ethanol while vortexing. Cells were fixed for 1 h at −20 °C, followed by two washing steps with PBS. The cell pellet was resuspended in PBS containing 1% FCS and 4 µL conjugated Ki-67-APC antibody (Biolegend) and stained for 30 min at room temperature. After washing, cells were resuspended in PBS for flow cytometric analysis. For PI staining, washed cells were resuspended in 1% FCS in PBS and cold 100% ethanol was added to the cells while vortexing. Fixation was carried out for at least 1 h at 4 °C. Fixed cells were centrifuged at 350 g for five minutes and resuspended in PBS containing 1% FCS, 0.1 mg/mL PI (Sigma-Aldrich, St. Louis, MO, USA) and 0.1 mg/mL RNAseA (ThermoFisher Scientific, Vilnius, Lithuania). Staining was carried out for 1 h at 37 °C in the dark. Subsequently, flow cytometric cell cycle analysis was performed using a Canto^TM^ II FACS device.

### 2.9. Colony Formation Assay

MaMel63a cells were non-adherently co-cultured with eosinophils in a 1:7.5 ratio. After 48 h, MaMel63a cells were counted and seeded on a 6-well plate for an additional 48 h. Cells were stained with crystal violet to visualize cell density. Cells were washed once with PBS and 0.25% crystal violet solution (Roth, Karlsruhe, Germany) (containing 20% methanol) was added. Wells were washed three times with deionized H_2_O and plates dried overnight. Representative images were taken. For quantification, equal amounts of methanol was added to each well. After 20 min of incubation, triplicates were transferred to a 96-well plate (Greiner Bio-One). Absorbance of crystal violet dye was measured at 570 nm using a Tecan Reader (Infinite M Nano).

### 2.10. ELISA

To determine ECP levels in the serum of late-stage melanoma patients and healthy volunteers, freshly drawn blood was centrifuged (800 g, 10 min, room temperature), serum aliquoted and stored at −80 °C until use. After thawing, samples were centrifuged for 15 min at 1000× *g* and ECP was measured by ELISA (Cusabio #CSB-E11729h). According to the manufacturer, the detection range was between 1.56 ng/mL and 100 ng/mL. Duplicates of each sample were assessed. Serum levels were correlated with the patient’s response to targeted therapy. Responders were defined as CR and PR at time of assessment and non-responders as PD and SD according to RECIST 1.1. Absorbance was measured at 450 nm using a Tecan Reader (Infinite M Nano).

### 2.11. Multiplex-Analysis

Customized human panels were used for LegendPlex^TM^ multi-analyte analysis (Biolegend) including eosinophil-related soluble mediators such as RANTES, sRAGE, Eotaxin, GM-CSF and APRIL. Serum from melanoma patients before and during administration of targeted therapy as a first-line therapy, were centrifuged before further processing. Serum analysis was carried out as described in the protocol provided by the manufacturer. Pre- and on-treatment concentrations of appropriate analytes were correlated with clinical response.

### 2.12. Statistical Analysis

Comparison of soluble factors and experimental data from melanoma patients prior to and during drug administration were analyzed using paired and unpaired t-tests, or a Mann–Whitney U test when normal distribution did not apply. Relative eosinophil counts (REC) of responders and non-responders were compared applying the Mann–Whitney U test. Analysis of in vitro cytotoxicity and assessment of eosinophil phenotype was performed using ANOVA with Bonferroni correction for three or more unmatched groups. Paired and unpaired t-tests were applied for two group comparisons. Prism (Graph-Pad, version 7) and/or SPSS (IBM, version 28.0) were used for visualizing the data.

## 3. Results

### 3.1. Patients

In total, data of 206 melanoma patients were used for correlative studies of peripheral eosinophil counts and eosinophil-secreted markers in response to treatment. The median age was 71 years, 97 patients were male (47.1%). Fifteen patients had unresectable stage III disease. The remaining 191 patients were assigned to the categories M1a (11.2%), M1b (24.8%), M1c (36.4%) and M1d (20.4%) according to the AJCC classification 2017 [51]. Time from pre-treatment blood collection to therapy commencement was 0–63 days. Sixty-seven percent of the patients included in the ECP analysis experienced an objective response (CR and PR). A BRAF mutation was detected in all patients receiving dual MAPKi. A detailed list of patient’s characteristics is presented in Table 1.

### 3.2. High Eosinophil Count Is Associated with Better Response to Targeted Therapy 

The relevance of eosinophil counts and ECP as a prognostic biomarker was evaluated in a cohort of 52 melanoma patients treated with first-line MAPKi. Established biomarker and peripheral blood counts were used for comparative analysis between responders and non-responders (Figure 1A,B). As expected, pre-treatment LDH was significantly higher in non-responders than in responders (*p* = 0.005) (Figure 1A). During treatment, responders are characterized by significantly lower LDH (*p* = 0.01) values compared to non-responders (Figure 1B). With respect to granulocyte counts, significantly higher absolute and higher relative eosinophil counts were detected pre- (AEC *p* = 0.0008, REC *p* = 0.05) and on-treatment (AEC *p* = 0.01, REC *p* = 0.008) in responders compared to non-responders (Figure 1A,B, Appendix A). In contrast, the on-treatment absolute neutrophil counts (ANC) tended to be higher in non-responders compared to the reciprocal group (*p* = 0.19). Comparing pre- and on-treatment samples, responders showed a significant decrease in absolute and relative neutrophil counts and a trend towards a decrease in absolute leukocyte counts during the course of treatment (Appendix A). High pre-treatment ECP levels correlated by trend with non-response to MAPKi (*p* = 0.12) (Figure 1C, left). On-treatment ECP showed no such association (*p* = 0.59) (Figure 1C, right). ECP values of healthy donors ranged from 2.5 ng/mL to 22.3 ng/mL (median 14.7 ng/mL) (data not shown). Additional eosinophil-associated soluble factors were measured in the sera of these patients. Pre-treatment proliferation-inducing ligand (APRIL) and on-treatment eotaxin-1 concentrations tended to be higher in non-responders (*p* = 0.09 and 0.19, respectively) (Appendix A). Interestingly, when measuring serum ECP in patients receiving immunotherapy, responders significantly show higher pre-treatment ECP concentration (*p* = 0.01) compared to non-responders. This observation was also seen during drug administration, with responders showing higher ECP concentration (*p* = 0.07) compared to non-responders (Appendix A). Interestingly, a high pre-treatment relative eosinophil count revealed a significant association with response to MAPKi (*p* = 0.013), but not anti-PD-1 based immunotherapy (*p* = 0.92) (Figure 1D).

### 3.3. Patient-Derived Eosinophils Show a Comparable Phenotype to Eosinophils from Healthy Donors

In order to identify potential similarities and differences in eosinophil phenotypes comparing patient-derived eosinophils with those from healthy donors, we analyzed twelve previously described surface markers on eosinophils from pre-treatment stage IV melanoma patients, healthy donors and stage I and II melanoma patients with no current evidence of disease (Table 2). The investigated molecules included activation (CD69, CD66b) and differentiation markers (Siglec-8), class I and II MHC proteins (HLA-A,-B,-C, HLA-DR), adhesion molecules (CD49d, CD29, CD31) and immunoregulatory receptors (TNFR2, CCR3, PD-L1) [52,53,54]. A representative gating strategy is shown in Figure 2A. The observed phenotypes were similar in all groups. Phenotypically, eosinophils from melanoma patients were comparable to the control cohort (stage I/II patients and HD). No significant difference was detected in any of the analyzed cell subsets. The median fluorescence intensity (MFI) for the targeted epitopes on eosinophils (defined as CD16-/CD66b+/CCR3 + cells) are shown in Figure 2B. Data for Siglec-8, pan-HLA, HLA-DR, TNFR2 and PD-L1 are not shown. 

### 3.4. Susceptibility of Melanoma Cells to Eosinophil-Mediated Killing Is Volatile 

As a high blood eosinophil count is associated with a better response to targeted therapy, we assumed that one of the driving mechanisms might be eosinophil-mediated cytotoxic activity against melanoma cells. We analyzed their anti-tumor capacity using different melanoma cell lines. Eosinophils displayed moderate cytotoxic activity against MaMel63a cells, a cell line originating from a skin metastasis carrying an activating BRAF mutation, thus providing a suitable model for further analyses. MaMel63a cells were cultured in the absence or presence of eosinophils with different tumor (T) to effector cell (E) ratios for 24 h. Eosinophils significantly decreased melanoma cell viability in a dose-dependent manner. Although a ratio of 1:5 was sufficient to reduce MaMel63a cell viability significantly, a ratio of 1:7.5 was superior in suppressing tumor viability and a 1:10 ratio showed no additional enhancement of the eosinophils’ cytotoxic effect (Figure 3A). Figure 3B shows a representative gating strategy quantifying the viability of melanoma cells after co-culture. A ratio of 1:7.5 T:E was used in further analyses. It has to be assumed that the tumoricidal function of eosinophils might vary depending on the donor and on the susceptibility of the melanoma cells. To assess the variation of the tumoricidal effect of eosinophils, we exposed three additional BRAF-mutated melanoma cells lines, MaMel80a, MaMel51 and MaMel06, and two non-melanoma cancer cell lines, H460 and WaGa, to eosinophils for 24 h under non-adherent culture conditions (Figure 3C, Appendix A). When different melanoma cells were exposed to eosinophils of the same donor, MaMel06 appeared to be the most susceptible cell line to eosinophil exposure as shown by a significant decrease in viability (mean drop in viability: 42.4%) after 24 h, followed by MaMel63a (mean drop in viability: 23.7%) and MaMel80a (mean drop in viability: 21.0%) when compared to the viability of the controls (melanoma cells alone). The MaMel51 cell line seemed to be the least affected by eosinophils (mean drop in viability: 12.9%) (Figure 3C). Eosinophils significantly induce early apoptosis and necrosis in MaMel63a cells and MaMel06 cells when cultured in a ratio of 1:7.5 for 24 h under non-adherent conditions. Under these conditions, necrosis was also induced in MaMel51 cells. None of the tested cell lines showed significant changes in the percentage of cells in late apoptosis upon exposure to eosinophils (Appendix A–C). As for the non-melanoma cell lines, H460 cells underwent apoptosis upon exposure to eosinophils at a ratio of 1:7.5 (Appendix A). Interestingly, the viability of the Merkel cell carcinoma cell line WaGa was efficiently diminished already at a 1:1 target to eosinophil ratio (Appendix A). Both H460 and WaGa show significant sensitivity to cisplatin treatment but not to vemurafenib and cobimetinib after 24 h (Appendix A). Examining the viability of cancer cell lines co-cultured with eosinophils, we observed a considerable variation in the extent of tumor cell apoptosis.

### 3.5. Melanoma Cells Affect Eosinophil Viability and Antigen Expression In Vitro

Having assessed the effects of eosinophils on melanoma cell viability, we wondered whether there might be a bidirectional relationship between both cell types. Therefore, we first analyzed eosinophil viability after adherent and non-adherent culture and evaluated whether exposure to MAPKi might change eosinophil function in co-culture with melanoma cells. Eosinophil viability turned out to significantly decrease after 24 and 48 h of non-adherent culture but was not affected by the presence of vemurafenib and cobimetinib (Figure 4A). Melanoma cells significantly increased eosinophil viability after 48 h but not after 24 h of co-culture. Interestingly, treatment with vemurafenib and cobimetinib significantly diminished the favorable effect of melanoma cells on eosinophil viability after 48 h of culture under non-adherent conditions. Whether the prolonged survival of eosinophils is unique to non-adherent cultures was tested by performing co-cultures in a 24-well plate to provide adherence. Melanoma cells were found to enhance viability of eosinophils after 48 h of co-culture under both non-adherent and adherent conditions (Figure 4B). 

To evaluate whether melanoma cells affect the expression of eosinophil surface molecules during co-culture, we phenotypically characterized healthy donor-derived eosinophils pre- and post-co-culture. Expression of CD69, an early activation marker, HLA-DR, PD-L1 and TNFR2 increased in eosinophils cultured for 48 h, relative to the MFI of controls (here: E 0 h) (Figure 4C). Co-culture with MaMel63a cells prevented an increase in the expression of the above mentioned antigens after 48 h. Interestingly, eosinophils express low levels of CD16. Culture for 48 h significantly increased the expression of CD16. Enhanced expression of this marker on eosinophils was significantly inhibited in co-cultures with MaMel63a cells. On the other hand, CD66b and CCR3 expression decreased during culture after 48 h relative to the control (here: E 0 h) (Figure 4C). Decreased expression of CD66b after 48 h of culture was significantly prevented upon co-culture with MaMel63a cells. Expression of the adhesion molecule CD31 and HLA-A, -B, -C on eosinophils decreased after 48 h of culture. MaMel63a cells did not show any effect on their expression.

### 3.6. Eosinophils from Peripheral Blood Donor-Dependently Reduce Melanoma Cell Viability 

Next, we investigated the cytotoxicity of eosinophils against melanoma cells to mechanistically link high eosinophil counts with prolonged survival of patients. In order to examine the functional activity of eosinophils derived from stage IV melanoma patients (Table 2), cytotoxicity against MaMel63a was measured pre- and on-treatment at week 6, 12, 24 and 48 (Figure 5). Eosinophils from healthy donors served as the control (Figure 5A). Viability of MaMel63a cells after 24 h of culture with and without eosinophils was inversely plotted to display eosinophil-mediated cytotoxicity. A cytotoxicity of one means melanoma cell viability was not affected by eosinophils and a cytotoxicity of two reports a reduction in melanoma cell viability by half compared to melanoma cells not exposed to eosinophils. A wide range of donor-dependent cytotoxicity by eosinophils (range 1.01–2.32) was recorded both in cells obtained from patients prior to treatment, as well as in the control group (range 1.04–2.19) (Figure 5A,B). A similar range was observed when addressing cytotoxicity in MaMel51 and MaMel80a (data not shown). Over the time of assessment during therapy, no trend regarding an increase or decrease in cytotoxicity was detected.

### 3.7. Tumoricidal Effect of Eosinophils Depends on Culture Conditions and It Is an Active Process Requiring Cell–Cell Contact

To unravel the underlying mechanism of melanoma cell–eosinophil interactions, we initially performed co-cultures under non-adherent culture conditions, forcing an interaction of melanoma cell lines with their surrounding co-cultured cells. By allowing melanoma cells to adhere to a given surface, the reduction in MaMel63a cell viability by eosinophils after 24 h was not measureable anymore (Figure 6A). Whether cytotoxicity is mediated by receptor engagement or soluble factors was determined by transwell experiments, where cultured eosinophils and melanoma cells were separated by a semipermeable membrane. Eosinophils were not capable of executing their cytotoxic function towards melanoma cells, suggesting the requirement of close proximity of eosinophils to their target (Figure 6B). This result was confirmed when exposing melanoma cells to a conditioned medium containing potential soluble factors from eosinophil (co-)cultures, which showed no effect on viability (data not shown). Interestingly, eosinophils enclose melanoma cells and form aggregates in both the absence and presence of vemurafenib and cobimetinib cultures after 24 h and 48 h (Figure 6C; Appendix A). However, eosinophils also formed aggregates in the absence of other cells in a medium containing vemurafenib and cobimetinib after 24 and 48 h of culture (Appendix A). In the absence of MAPKi, these aggregates were only seen when incubating for 48 h, but not after 24 h (Appendix A). Additionally, we tested various surface molecules that have been previously suggested as potential candidates for eosinophil–melanoma cell interaction, such as CD11b, CD11a, CD18, CD54, CD49d and VCAM-1 [27,55]. None of these appeared to be involved in eosinophil cytotoxicity, since blocking antibodies against these receptors showed no effect (data not shown). In order to verify whether the observed tumoricidal effect relies on an active process, MaMel63a cells were co-cultured either with viable, lysed or heat-inactivated lysed eosinophils, preventing active interaction. Compared to viable eosinophils, cell lysates further reduced MaMel63a cell viability significantly. Neutralization of lysed eosinophil content through heat-inactivation led to full abrogation of cytotoxicity towards MaMel63a cells (Figure 6D). To provide evidence regarding the effect of eosinophils on the cell cycle and proliferation of melanoma cells, we conducted propidium-iodide-based (PI) and Ki-67-specific stainings and performed colony formation assays to evaluate surviving melanoma cell numbers upon co-culture. Eosinophils showed no significant impact on either readout. Proliferation, the cell cycle and the capability of melanoma cells to spread on a given surface were not affected upon exposure to eosinophils (Figure 6E–G).

### 3.8. Tumoricidal Effects of Eosinophils and Systemic Therapy Are Additively Cytotoxic to Melanoma Cells

Since we could not identify a soluble factor that is suitable for the prognostic evaluation of melanoma patients receiving targeted therapy, we next analyzed how treatment in combination with eosinophils affects melanoma cell viability in vitro. MaMel63a cells were non-adherently cultured in a medium containing vemurafenib and/or cobimetinib with or without eosinophils. The viability of MaMel63a cells and eosinophils was measured after 24 and 48 h (Figure 7). Interestingly, the presence of BRAF/MEK-inhibitors, both as single or combined treatment, significantly enhanced the eosinophil-mediated cytotoxicity towards MaMel63a cells after 24 h (Figure 7A). The additional drop in MaMel63a cell viability was also observed after 48 h when using only vemurafenib and eosinophils (Figure 7B). An additive reduction in viability using cobimetinib only or combination therapy with vemurafenib, after 48 h, was not detectable.

## 4. Discussion

In order to optimize treatment efficacy, reliable and easily accessible biomarkers are required. In this study, we provide data from a homogeneous cohort investigating the relevance of eosinophils in late-stage melanoma patients receiving first-line targeted therapy. Additionally, we investigated the function and phenotype of eosinophils to reveal their prognostic relevance in metastatic melanoma. Thereby, we attempted to clarify how high blood eosinophil counts and eosinophil toxic granules contribute to tumor rejection and are thus a biomarker candidate for overall survival and response of melanoma patients to targeted therapy. Despite the apparent beneficial effect of eosinophils in melanoma, studies also point at a link between eosinophil count and the occurrence of immune-related adverse events (irAEs) during treatment with immune checkpoint inhibitors [56]. As personalized therapies are on the rise, there is a need for models predicting risk–benefit ratios instead of single clinical endpoints [57]. However, data on treatment-related adverse events were not collected by us. In line with previous observations, we showed an association of high AEC and REC, as well as low LDH with response to MAPKi pre- and on-treatment [39,41]. Importantly, we could show a robust association of high pre-treatment relative eosinophil counts with the response to targeted therapy. Previous studies reported drastic changes in AEC upon exposure to immunotherapy in melanoma patients, observable on treatment [47]. Our data now adds REC prior to MAPKi to the list of biomarkers predicting the outcome before commencing therapy, a finding more applicable than on-treatment changes. 

Direct eosinophil cytotoxicity towards melanoma cells might explain the beneficial influence of blood eosinophils on a patient’s outcome. We demonstrated that freshly isolated blood-derived eosinophils were able to robustly induce melanoma cell apoptosis in different melanoma and non-melanoma cell lines. Additionally, we could show that cytotoxicity depends on the target to effector cell ratio, as more eosinophils are added to the co-culture, a higher degree of apoptosis is observable in the melanoma cells. An association between cytotoxicity and clinical outcome, however, could not be shown. Studies report the requirement of CD11a/CD18 (LFA-1) for eosinophils to exert their cytotoxic function towards colon carcinoma [25]. Neutralization of CD18 abolished the beneficial inhibition of tumor growth in vitro in Colo-205 and in the colons of mice [18,58]. We explored a variety of suggested receptors, including LFA-1, that might mediate the close interaction of eosinophils and melanoma cells, but none of the tested blocking antibodies was sufficient to prevent cytotoxicity towards melanoma cells. However, exploring the interaction of eosinophils and melanoma cells, we could show that eosinophil-induced apoptosis in melanoma depends on the close cell–cell proximity and could be induced neither by separate co-culture nor by eosinophil conditioned medium. This observation is in line with the cytotoxicity of blood eosinophils towards a colon carcinoma cell line, which showed the necessity of the direct contact of eosinophils to their target to induce apoptosis [18]. To define melanoma cell interaction, initial co-cultures were performed under non-adherent conditions. This imitates the interaction of circulating tumor cells (CTCs) with peripheral blood eosinophils. Melanomas exhibit high metastatic potential with frequent occurrence of CTCs in the bloodstream [59,60,61]. Interestingly, activated eosinophils show robust inhibition of pulmonary metastasis and enhance tumor rejection by improving T cell infiltration in melanoma mouse models [5,50,62,63]. It might be possible that CTCs are exposed to eosinophils and activate the bloodstream eosinophils killing mechanism, which might explain the beneficial effect of high circulating eosinophil counts in melanoma patients. This observation is supported by our in vitro experiments showing that eosinophils, in combination with targeted therapy, significantly and additively reduce melanoma cell viability under non-adherent conditions. Thus, there appears to be a beneficial cooperation between eosinophils and MAPKi for the treatment of patients with advanced melanoma. Interestingly, MAPKi induced aggregation of eosinophils alone after 24 h. It is possible that MAPKi induces degranulation and/or adhesion in eosinophils and might lead to the expression of an “eat-me” signal on melanoma cells. Eosinophils show versatile effects in cancer, depending on the location and given conditions [64]. Changing from non-adherent cultures to adherent conditions, rather mimicking an environment encountered in solid tumors, resulted in eosinophils that were unable to exert an apoptosis-inducing function towards melanoma cells. As for melanoma cells, exposure to different surface structures influences their adherence properties and may change survival signaling for resistance against eosinophil-mediated cytotoxicity [65]. Screening for various eosinophil markers, we can show that pre-treatment blood eosinophils of late-stage melanoma patients and controls show comparable expression patterns. Activation markers like CD69 and CD66b and adhesion molecules like CD29 and CD31 are similarly expressed in the compared cohorts. Without ruling out potential differences in other markers that were not assessed in this study, we assume that phenotypical changes and differences in eosinophils might happen during treatment and/or during direct contact with melanoma cells. The latter is supported by the downregulation of CD69 and upregulation of CD66b after in vitro co-cultures. Since we could also show that eosinophils are significantly more viable when co-cultured with melanoma cells, there is a bidirectional interaction of the cells. CD66b was described to promote cellular adhesion of eosinophils and expression can be upregulated in vitro by IL-5 [66]. IL-5 or chemoattractants such as GM-CSF are also able to induce expression of CD69, CD16 and HLA-DR in vitro [67,68,69]. The upregulation of CD66b measured might explain the ability of eosinophils and melanoma cells to form aggregates, which we have observed. Melanoma cells seem to prevent potential inhibition of metastases by regulating the activation of eosinophils, while allowing and regulating adhesion and improving the survival of eosinophils in vitro. 

A melanoma mouse model revealed the formation of stable aggregations of eosinophils and cancer cells via IL-33 stimulation [27]. In another melanoma mouse model, eosinophil accumulation around a solid tumor was reported to be partially restricted to the necrotic and capsule regions [70]. There are several reported mechanisms describing degranulation [71], one of which involves cytolysis [72]. Enhanced eosinophil viability in co-culture with melanoma cells might be caused by immune synapse-like mechanism. Andreone et al. proposed such a mechanism was involved in the interaction of eosinophils and tumor cells [27]. Aggregation of these two cell subtypes potentially stimulates the production of GM-CSF by melanoma cells. GM-CSF is produced by a variety of cells including epithelial cells and numerous types of cancer, and is able to prolong the survival of eosinophils in a co-culture [73,74,75,76]. This positive effect on eosinophil viability was also seen in other studies in co-cultures with glioblastoma multiforme or conjunctival fibroblasts [77,78]. We could show that induced apoptosis in melanoma cells is mediated by intact eosinophils and is an active process in vitro. The exposure to released eosinophil content by lysis induced enhanced melanoma cell destruction, which could be prevented by heat-inactivation. This observation is in accordance with a study showing the toxicity of eosinophil content towards B16 melanoma cells [50]. Taken together, our data show that melanoma cells exert a regulatory function towards eosinophils and their interaction is an active process. Additionally, we hypothesize that the release of stress signals by melanoma cells due to the exposure to MAPKi might enhance the efficacy of eosinophils to induce apoptosis in melanoma cells, which might explain the additive cytotoxic effect of eosinophils and MAPKi in vitro [70,79]. Besides studying the phenotype and function of eosinophils, we investigated pre-treatment serum ECP concentrations in patients receiving targeted therapy as a first-line treatment. ECP, a toxic granule protein secreted by eosinophils, is cytotoxic towards cancer cells in vitro [80] and has been previously proposed to be a potential prognostic marker for malignant melanoma [16]. Krückel et al. showed that higher ECP values are associated with a greater risk of disease progression. In addition, serum ECP correlated inversely with overall survival in a heterogeneous study cohort [16]. In our cohort, we found a trend towards lower levels of serum ECP in patients responding to MAPKi, while responders to monotherapy with PD-1 blockade and combination therapy with CTLA-4-blockade had significantly higher levels than non-responders. Considering our and other data together, ECP seems to be a predictive biomarker that might help to choose first-line therapy.

## 5. Conclusions

Taken together, our data indicate that high eosinophil counts prior to MAPKi initiation in patients with metastatic melanoma, but not ECP serum concentrations or functional differences in eosinophils, are associated with their response to targeted therapy. Our results are consistent with previous studies linking eosinophil blood counts to better survival in melanoma patients. Functional assays revealed a close and active interaction of peripheral eosinophils and melanoma cells in vitro. The interaction with melanoma cells seems bidirectional and depends on various factors like the donor, providence of adherence and the type of melanoma. Eosinophils significantly induce apoptosis in melanoma cells, which could be reinforced by additional treatment with BRAF-/MEK-inhibitors. The mechanism of this additive effect still needs to be unraveled, but our data suggest that eosinophils are a potential prognostic biomarker that warrants further studies. In addition, we provide first insights into the complex and treatment-dependent role of eosinophils and their secreted molecules in melanoma. 

## Figures and Tables

**Figure 1 cancers-14-02294-f001:**
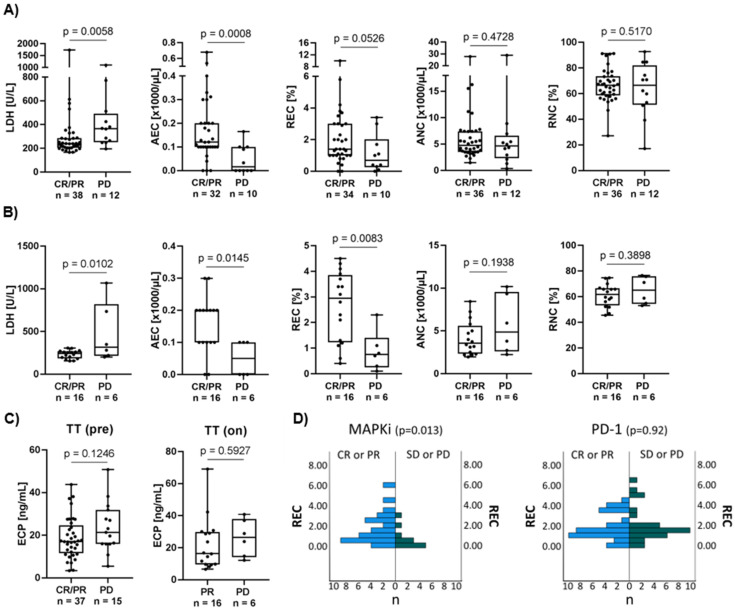
Association of serum ECP with therapy response and clinically relevant prognostic markers. (**A**,**B**) Association of response to targeted therapy with established prognostic markers such as LDH, AEC, REC, ANC and RNC. Comparing depicted clinical blood parameters of responders to non-responders (**A**) prior to treatment and (**B**) during drug administration. Responders show significantly higher AEC values prior to administration compared to non-responders. Responders are defined as CR or PR and non-responders as PD at time of assessment. Box plots show levels of clinical markers (median and the 25th and 75th percentiles; whiskers represent minimal and maximal outliers), as well as individual data points. (**C**) Comparison of serum ECP concentration (ng/mL) of responders and non-responders (**left**) prior (TT pre) and (**right**) during (TT on) targeted therapy. There is a trend towards higher pre-therapeutic values of ECP in melanoma patients with disease progression compared to responders. (**D**) Association of pre-treatment REC with response to MAPKi or PD-1 treatment in an independent patient cohort. In total, 50 patients (CR or PR *n* = 39; SD or PD *n* = 11) with metastatic melanoma were included in the MAPKi cohort and 59 patients (CR or PR *n* = 32; SD or PD *n* = 27) were included in the PD-1 cohort. High REC was significantly correlated with better response to MAPKi but not to PD-1 monotherapy. Mann–Whitney U test was used to compare REC in responders and non-responders.

**Figure 2 cancers-14-02294-f002:**
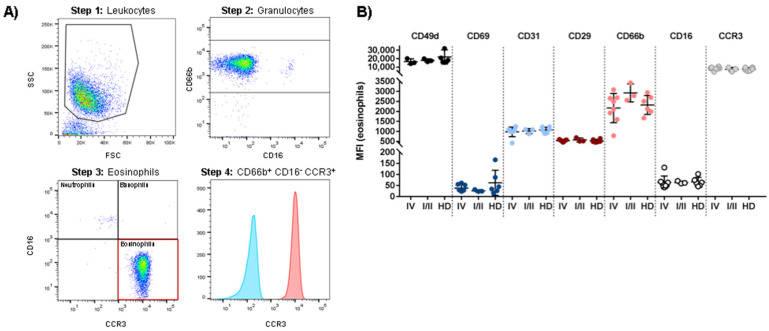
Phenotypic characterization of peripheral blood eosinophils of stage IV and stage I/II melanoma patients compared to healthy donors (HD). (**A**) Dot plots and histogram identifying freshly isolated CD66b+ CD16- CCR3+ eosinophils. To remove doublets and dead cells, cells were gated for singlets (not shown), followed by gating on forward (FSC) and side scatter (SSC) (Step 1). Granulocytes were identified as CD66b-positive cells (Step 2). Next, the granulocyte subpopulation was subdivided (Step 3). Eosinophils were defined as CD16- CCR3+ cells. In the last step, the expression of the target epitope (here: CCR3) on eosinophils was shown in a histogram compared to the unstained eosinophil sample (Step 4). (**B**) MFI determination for the expression of CD49d, CD69, CD31, CD29, CD66b, CD16 and CCR3 (CD193) within freshly isolated eosinophils. Each dot represents an individual donor. Analysis included a total of 13 patients with advanced melanoma, six patients with early stage melanoma and 12 healthy donors.

**Figure 3 cancers-14-02294-f003:**
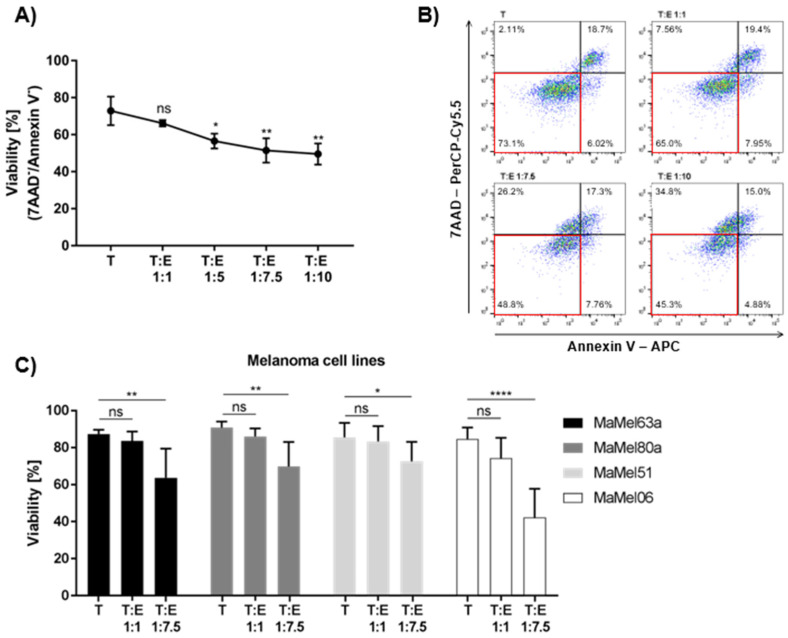
Eosinophils exert tumoricidal function towards various melanoma cell lines to different extents. Non-adherent (co-)cultures of melanoma cell lines (T) with or without eosinophils (E) for 24 h. (**A**) Eosinophils show a significant target to eosinophil ratio-dependent cytotoxic effect towards MaMel63a cells. A ratio of 1:5 melanoma cells to eosinophils is sufficient to significantly decrease MaMel63a cell viability. Mean percentage of MaMel63a cell viability is shown for three independent experiments. (**B**) Representative viability gating strategy for 7-AAD / Annexin-V staining of 24 h co-cultures of MaMel63a cells with eosinophils. Viable cells displayed in red box. (**C**) Cytotoxicity assessed in different melanoma cell lines, MaMel63a, MaMel80a, MaMel51 and MaMel06. Mean percentage of melanoma cell viability ± standard deviation (SD) is shown for six to eight independent experiments. ns *p* > 0.05, * *p* < 0.05, ** *p* < 0.01, **** *p* < 0.0001.

**Figure 4 cancers-14-02294-f004:**
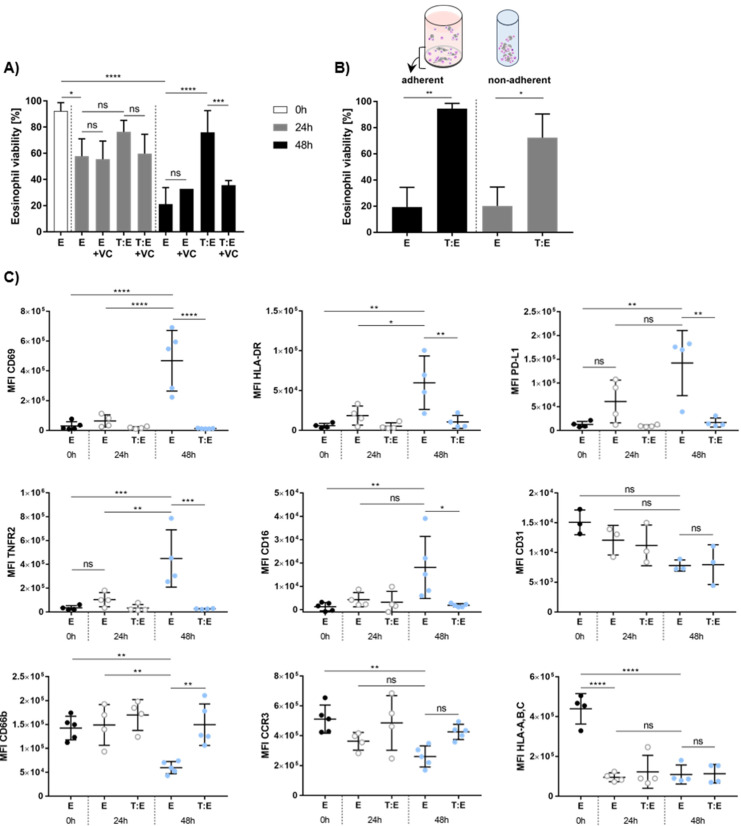
In vitro viability and phenotypic characterization of eosinophils in co-culture with melanoma cells. (**A**) Eosinophil viability increases upon co-culture with melanoma cells. Non-adherent (co-)culture of eosinophils and MaMel63a cells in medium or medium containing 1 µM vemurafenib and 100 nM cobimetinib for 24 and 48 h. Mean percentage of eosinophil viability ± standard deviation (SD) shown for one to nine independent experiments. (**B**) MaMel63a cells were co-cultured with eosinophils in a 1:7.5 tumor to effector cell ratio for 48 h. Separate measurement of viability of adherent (24-well plate) and non-adherent (polypropylene tube) eosinophils. Eosinophils in co-cultures with melanoma cells are significantly more viable. Mean percentage of eosinophil viability ± SD is displayed from two to five independent experiments. (**C**) Phenotypic characterization of peripheral blood eosinophils from healthy donors in (co-)culture with or without MaMel63a cells. Phenotypic epitopes were analyzed at time points 0 h, 24 h and 48 h using flow cytometry. The MFI of the target epitopes on eosinophils were plotted and the expression levels at time point 0 h were compared to expression after 24 h and 48 h (co-)culture. MaMel63a cells appear to be regulating surface marker expression of eosinophils after 48 h co-culture. The MFI ± standard deviation (SD) is shown from three to five independent experiments. ns *p* > 0.05, * *p* < 0.05, ** *p* < 0.01, *** *p* < 0.001, **** *p* < 0.0001.

**Figure 5 cancers-14-02294-f005:**
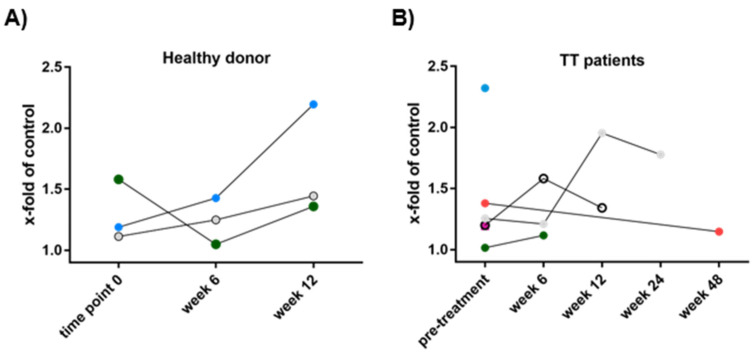
Sequential assessment of cytotoxicity by peripheral blood eosinophils towards melanoma cells. Eosinophils from melanoma patients were isolated prior to treatment and during therapy at weeks 6, 12, 24 and 48. Co-culture of MaMel63a cells and eosinophils serially obtained from (**A**) healthy donors (HD) or (**B**) stage IV melanoma patients subsequently receiving targeted therapy (TT) in a 1:7.5 ratio, non-adherent for 24 h. In order to display eosinophil-induced MaMel63a cell death, viability of MaMel63a cells relative to controls was plotted inversely. A high number refers to a high cytotoxic eosinophilic activity. Each symbol represents a healthy donor or patient, respectively. Eosinophil-mediated cytotoxicity towards MaMel63a cells is donor-dependent. *n* = three healthy donors, *n* = six patient samples obtained prior to melanoma treatment.

**Figure 6 cancers-14-02294-f006:**
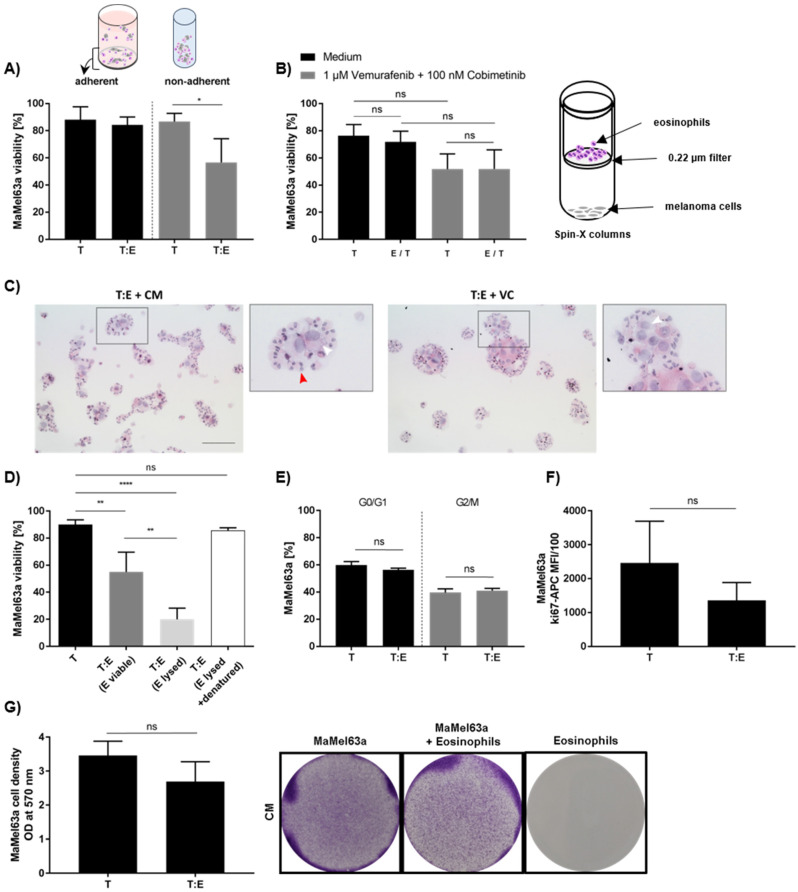
Unraveling the interaction of melanoma cells and blood-derived eosinophils. (**A**) MaMel63a cells were co-cultured with eosinophils for 48 h. Separate measurement of viability of adherent (24-well plate) and non-adherent (polypropylene tube) melanoma cells. Cytotoxicity depends on culture condition. Melanoma cell killing by eosinophils is only observed under unattached conditions. (**B**) Physical proximity is crucial for the functionality of eosinophils in co-cultures. Separate co-cultures of MaMel63a cells with eosinophils in Spin-X^®^ columns containing a semipermeable membrane (pore size 0.22 μm) for 24 h. Culture with or without 1 μM vemurafenib and 100 nM cobimetinib. The cytotoxic effect could not be maintained when cells were cultured separately. (**C**) Cytospins of co-cultures stained for HE. Eosinophils and MaMel63a cells form aggregates after 24 h of culture. Red arrows indicate intact eosinophils. Black square in original image indicate the magnified (3.2×) image. The white arrows indicate melanoma cells. 100 µm scale bar. (**D**) Tumoricidal function of eosinophils towards melanoma cells is an active process. Preventing signal transduction in co-cultures by lysis (1 min liquid nitrogen treatment of eosinophils and subsequent freeze–thaw) and neutralization (heat-inactivation for 1 h at 95 °C after lysis) of eosinophil content prior culture. Non-adherent co-cultures of MaMel63a cells with or without viable, lysed or neutralized eosinophils for 48 h. Significant enhancement of cytotoxicity towards MaMel63a cells was observed when cultured with lysed eosinophils. Heat-inactivation of eosinophil content abrogates cytotoxicity towards MaMel63a cells. (**A**,**B**,**D**) Mean percentage of melanoma cell viability ± SD is displayed from three to four independent experiments. (**E**,**F**) Examining cell cycle and proliferation in MaMel63a cells exposed to eosinophils. PI staining and Ki-67-specific staining of MaMel63a cells in co-cultures with eosinophils after 48 h. Eosinophils do not affect MaMel63a cell cycle nor proliferation. Mean percentage of MaMel63a cells (**E**) in G0/G1 and G2/M phase and (**F**) positive for Ki-67 is displayed from three independent experiments. (**G**) Assessment of melanoma cell survival using colony formation assay in co-cultures. MaMel63a cells were non-adherently co-cultured with eosinophils for 48 h. MaMel63a cells were counted and seeded on 6-well plates for additional 48 h. Co-culture was subsequently stained with crystal violet to visualize cell density (**right**). The cell density was quantified measuring the absorbance of the crystal violet dye (**left**). Eosinophils do not affect cell density of MaMel63a cells. Quantitative crystal violet stainings are shown for three independent experiments. All presented co-cultures were performed using a 1:7.5 melanoma cell (T) to eosinophils (E) ratio. ns *p* > 0.05, * *p* < 0.05, ** *p* < 0.01, **** *p* < 0.0001.

**Figure 7 cancers-14-02294-f007:**
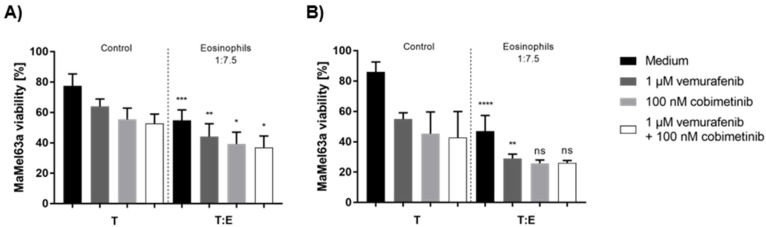
Additive suppressing effect of eosinophils and treatment on MaMel63a cell viability in BRAF/MEK-inhibitor-containing medium. Non-adherent (co-)culture of CFSE-stained MaMel63a cells with or without freshly isolated eosinophils at 1:7.5 ratio in medium with or without 1 µM vemurafenib and/or 0.1 µM cobimetinib for (**A**) 24 and (**B**) 48 h. Mean percentage of melanoma viability ± SD are shown from three to four independent experiments. Significances compared to respective controls (treated and untreated melanoma cells) without granulocytes are shown. ns *p* > 0.05, * *p* ≤ 0.05, ** *p* ≤ 0.01, *** *p* ≤ 0.001, **** *p* ≤ 0.0001.

**Table 1 cancers-14-02294-t001:** Patients included for evaluation of serum ECP, peripheral blood counts and association of REC with response to MAPKi in patients with advanced malignant melanoma receiving dual targeted therapy (total *n* = 94). Patients receiving immunotherapy served as control cohort (total *n* = 112).

Variables		Patients	
**Age**	median (range)	71 years (27–93)	
			%
**Individual Patients**		206	100
Sex	male	97	47.1
	female	79	38.3
	unknown ^1^	30	14.6
Stage	III	15	7.3
	IV	191	92.7
M-Category	M1a	23	11.2
	M1b	51	24.8
	M1c	75	36.4
	M1d	42	20.4
First-Line Therapy	yes	180	87.4
	no	26	12.6
Therapy after Study Inclusion	anti-PD-1	86	41.8
	anit-PD-1 + anti-CTLA-4	26	12.6
	BRAFi + MEKi ^2^	94	45.6
LDH	>1× ULN ^3^	68	33.0
	<1× ULN ^3^	134	65.1
	missing	4	1.9

^1^ For patient serum samples from Tübingen/Erlangen, this information was not provided. ^2^ All patients receiving dual MAPKi (BRAFi + MEKi) showed a BRAFV600-mutation. ^3^ Upper limit of normal (ULN).

**Table 2 cancers-14-02294-t002:** Patients and healthy donors included in phenotyping eosinophils and for cytotoxicity evaluation ^1^.

Variables			Donor	
				%
**Individual Donors**			31	100
Patient	Stage	I/II	6	19.4
		IV ^2^	13	41.9
Healthy Donor			12	38.7

^1^ For functional and phenotypic analyses, additional blood samples were collected from donors. Clinical parameters were not collected for this cohort. ^2^ Treatment-naïve stage IV patient samples.

## Data Availability

Anonymized patient and experimental data are made available by the senior author of this study upon request for academic studies.

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
