# Peer review of "Blood Eosinophils Are Associated with Efficacy of Targeted Therapy in Patients with Advanced Melanoma"

_cancers, 2022, doi:10.3390/cancers14092294_

Round 1

Reviewer 1 Report

Specific comments:

Major comments:

1) It would be easier for the reader to follow the text about and from the different cohorts if all cohorts, including the control cohort and with all subjects were presented in tables and that the text was clarified throughout. For instance, on line 126 about dual target therapy, which therapies did the patients referred to here receive? Who are the patients in figure 1 in relation to the text in the Methods section and Table 1?

2) Figure 2a: Please explain the gating better. Are you showing all steps? Does the top right graph show the cells from the gate in the graph to the left? But if so, what happened to the neutrophils (which presumably are in the scatter gate in the top left)? Why has the gate in top left been drawn to include some of the events in the bottom left (and what may those be, monocytes, lymphocytes, RBC, platelets…)?

3) Regarding the adherent conditions: Were wells or dishes coated with any adhesive, e.g., extracellular matrix, protein ligand (and blocked) to make cells adhere? For instance, eosinophils will adhere specifically to some proteins and not others, and unless the cells have been activated with, e.g., IL5, they will only adhere to VCAM-1. Other substrates such as ICAM-1, periostin, etc., require cell activation. See, e.g., J. Leukoc. Biol. 2008, 83:1-12, and other reviews. Or do you think that the eosinophils adhered to a factor, e.g., vitronectin or fibrinogen, present in the added serum (but that too would need cell activation)? What proportions of cells were usually adherent and non-adherent? If the cells don’t adhere specifically to a ligand (and the substrate not blocked), they may adhere just to the plastic or glass, which would be unphysiological.

4) Abstract, line 48: Where in the manuscript does this p value of 0.013 come from (this reviewer appears to have missed it)?

5) Fig. 4b: Do you care to speculate what the active factor(s) from tumor cells that enhances eosinophil viability may be? May it be a cytokine like GM-CSF and/or some direct cell-cell interaction?

Minor comments:

6) Fig. 1: The legend says that mean and SEM is shown, but it looks like the graphs show median, quartiles, and then perhaps 10th-90th percentiles. Please correct if needed.

7) Data in fig. 3c: This part presumably presents data on the viable cells in the red box in b, i.e., 7AAD-negative/annexin V-negative cells. Please also present data on the cells in the other quadrants, including apoptotic and necrotic cells.

8) As the data in fig. 4c are very interesting, it would be even more interesting if the authors related their data to what is known about which factors and under what circumstances these surface proteins on eosinophils are upregulated (see, e.g., Clin. Exp. Allergy 2014, 44:482-98).

9) Fig. 6b: Is there a significant difference between VC and medium?

10) Lines 158-160: Which antibodies were used in the Miltenyi kit?

11) Line 170: A purity of 90% seems low. It should be possible to achieve 98% purity on a regular basis with negative selection using multiple antibodies (as in, e.g., PLoS One 2018, 13:e0201320). What cells are the up to 10%?

12) Line 282, “assuming” normal distribution, where the distributions actually normal?

13) Lines 314-315: “ECP… …were correlated” refers to fig. 1c, but the p values in 1c are 0.12 and 0.59, which are not significant. As they are not, don’t write that they correlated. If mentioning at all, rewrite to “tended to correlate”, “there was a trend to a correlation”, or “there was a tendency to…”, etc. In consistency, please adjust the Discussion and Abstract if needed.

14) Similarly, on lines 320-322 the text refers to non-significant data and can be deleted or should be rewritten.

Author Response

Reviewer 1:

Major comments:

1) It would be easier for the reader to follow the text about and from the different cohorts if all cohorts, including the control cohort and with all subjects were presented in tables and that the text was clarified throughout. For instance, on line 126 about dual target therapy, which therapies did the patients referred to here receive? Who are the patients in figure 1 in relation to the text in the Methods section and Table 1?         
            Answer: We apologize for the lack of clarity. In hope to clarify the cohorts used, Table 2 was moved from the material and methods section to the respective result section in point 3.3. to which it is referring to. Table 2 was also linked to point 3.6.    
To evaluate eosinophil-related biomarkers in patients, cohort 1 (Table 1) was created with treatment-naïve and corresponding follow-up serum samples from staging time points. We aimed to collect as many data (lab values, ECP values etc.) as possible to increase the power of our results. 
As for evaluating eosinophil phenotype and in vitro cytotoxicity , we collected treatment-naïve stage IV melanoma patient, stage I/II and healthy donor blood samples for cohort 2 (Table 2) to isolated eosinophils and to which laboratory values were not required. In this perspective, only minimal data were collected since no associative studies were performed.

2) Figure 2a: Please explain the gating better. Are you showing all steps? Does the top right graph show the cells from the gate in the graph to the left? But if so, what happened to the neutrophils (which presumably are in the scatter gate in the top left)? Why has the gate in top left been drawn to include some of the events in the bottom left (and what may those be, monocytes, lymphocytes, RBC, platelets…)?           
            Answer: Thank you for this comment. We apologize for not showing all gating. We added the information and changed the legend of Figure 2. Additionally, we adjusted Figure 2A by numbering the gating steps. The amount of neutrophils (and other leucocytes) in this population is minimized as after blood draw, the eosinophil population is purified using the automatic magnetic labeling-based system autoMACS from Miltenyi Biotec. There is always some degree of neutrophil “left-overs”, which you can find in “Step 3” of Figure 2A in the upper left gating box. The cells in Figure 2A “Step 1” outside the gating box are mostly dead and unstained cells.

3) Regarding the adherent conditions: Were wells or dishes coated with any adhesive, e.g., extracellular matrix, protein ligand (and blocked) to make cells adhere? For instance, eosinophils will adhere specifically to some proteins and not others, and unless the cells have been activated with, e.g., IL5, they will only adhere to VCAM-1. Other substrates such as ICAM-1, periostin, etc., require cell activation. See, e.g., J. Leukoc. Biol. 2008, 83:1-12, and other reviews. Or do you think that the eosinophils adhered to a factor, e.g., vitronectin or fibrinogen, present in the added serum (but that too would need cell activation)? What proportions of cells were usually adherent and non-adherent? If the cells don’t adhere specifically to a ligand (and the substrate not blocked), they may adhere just to the plastic or glass, which would be unphysiological.         
            Answer: This is an important point made by the reviewer. Polystyrene wells were not coated with any adhesive. Melanoma cells were adherent in the well, as we could observe by the change of their morphology. Performing flushing experiments with CM, melanoma cells did not detach from the vessel, unless they were dead and already floating. As for eosinophils, we assumed that by not adding any additional adhesive factor to the culture e.g. VCAM-1 etc., we would force eosinophils to interact with melanoma cells, adhere and attack them in adherent culture, and reduce melanoma cell viability, which was disproven by our results. We did not count the proportion of attached/detached eosinophils in culture after 24 hours. However, when washing the wells to detach melanoma cells for viability measurements, the majority of eosinophils were removed as well. 

4) Abstract, line 48: Where in the manuscript does this p value of 0.013 come from (this reviewer appears to have missed it)?     
            Answer: The p value of 0.013, calculated using the Mann-Whitney-U test, can be found in Figure 1 D) showing the association of high pre-treatment relative eosinophil count (REC) with response to MAPKi. The corresponding text is found in the last lane of section 3.2. We added the respective information to the text.

5) Fig. 4b: Do you care to speculate what the active factor(s) from tumor cells that enhances eosinophil viability may be? May it be a cytokine like GM-CSF and/or some direct cell-cell interaction?
            Answer: Thank you very much for this comment. Indeed, it would be of great interest to find the driving factor that enhances eosinophil viability in co-culture with melanoma cells. We added a paragraph in the discussion section.            
There are several potential options. One of which might involve an immune synapse-like mechanism. We imagine following scenario: As our data show, eosinophil-melanoma cell co-culture initiates aggregation of these two cell subtypes in vitro (Figure 6 C), it might be that this rather close proximity generates an “immune synapse”, involving cell-cell contact and the release of cytokines. In this synapse, melanoma cells might be stimulated to produce GM-CSF, as you have mentioned and proposed as a potential mediator for improved survival of eosinophils in co-culture.  

Minor comments:

6) Fig. 1: The legend says that mean and SEM is shown, but it looks like the graphs show median, quartiles, and then perhaps 10th-90th percentiles. Please correct if needed.
            Answer: Thank you very much for this correction. The reviewer is right. Indeed, clinical markers are shown in box plots with median, the 25th and 75th percentiles and whiskers represent minimal and maximal outliers. The legend of Figure 1 was adjusted accordingly.

7) Data in fig. 3c: This part presumably presents data on the viable cells in the red box in b, i.e., 7AAD-negative/annexin V-negative cells. Please also present data on the cells in the other quadrants, including apoptotic and necrotic cells.         
            Answer: The reviewer is correct. Figure 3C presents the viability of melanoma cells (defined as 7-AAD- and Annexin V-negative cells). As requested, we added the information about the percentage of cells in early apoptosis, late apoptosis and necrotic cells in the supplementary materials. It is now shown as Figure S4 and was embedded into the text in section 3.4. The following supplementary figures were renumbered in the text to match text and figures. The total number of supplementary figures is now six.

8) As the data in fig. 4c are very interesting, it would be even more interesting if the authors related their data to what is known about which factors and under what circumstances these surface proteins on eosinophils are upregulated (see, e.g., Clin. Exp. Allergy 2014, 44:482-98).
            Answer: CD69, CD16 and HLA-DR expression were now briefly discussed in the manuscript. The suggested reference was cited and added to the reference list. 

9) Fig. 6b: Is there a significant difference between VC and medium?        
            Answer: Thank you for this question. The analysis showed that there is a trend (p = 0.0857) towards a decrease in MaMel63a cell viability when exposed to 1 µM vemurafenib and 100 nM cobimetinib in the Transwell experiment compared to non-treated MaMel63a cells. In the setup with separately kept eosinophils, MaMel63a viability was also tending to decrease (p = 0.1801). This Transwell experiment was performed three times. We assume that increasing the number of repetitions, we would reach significance. Figure 6 B) was changed accordingly and significances (here: ns) were added to requested comparison.

10) Lines 158-160: Which antibodies were used in the Miltenyi kit?     
 Answer: The missing information was added to the material and methods section to point 2.2.

11) Line 170: A purity of 90% seems low. It should be possible to achieve 98% purity on a regular basis with negative selection using multiple antibodies (as in, e.g., PLoS One 2018, 13:e0201320). What cells are the up to 10%?       
            Answer: Purifying eosinophils using the autoMACS (miltenyi biotec), we observed a wide range of purity after automatic labeling. The variety might be explained by the different batches of eosinophil antibody cocktail, the age of the labeling columns (as they can oxidize during time of use) and the sample itself. Additionally, we experienced lower eosinophil purity, when the sample even after RBC lysis was still containing some degree of erythrocytes “left-overs”. The remaining cells are neutrophils. We have used neutrophils as controls and in (co-)cultures, these granulocytes perform differently and non-toxic compared to eosinophils in the tested melanoma cells lines.  

12) Line 282, “assuming” normal distribution, where the distributions actually normal?
            Answer: Thank you for pointing it out. We apologize for not testing it earlier. We re-ran the analysis for Figure 1 and normal distribution of data was confirmed. We modified the material and method section in point “2.12. Statistical Analysis” accordingly. For pre-treatment LDH, AEC, REC and ANC values, Mann-Whitney test was now applied as data show no normal distribution. For on-treatment values, only AEC had to be changed to Mann-Whitney test. Additionally, the y-axis for on-treatment RNC was changed for consistency and for AEC and ANC concentration in square brackets µl was changed to µL. For consistency, Supplementary Figure S1A) was changed as well. Here, ALC data of non-responders were not normal distributed and Mann-Whitney test was applied. P-values was adjusted accordingly.

13) Lines 314-315: “ECP… …were correlated” refers to fig. 1c, but the p values in 1c are 0.12 and 0.59, which are not significant. As they are not, don’t write that they correlated. If mentioning at all, rewrite to “tended to correlate”, “there was a trend to a correlation”, or “there was a tendency to…”, etc. In consistency, please adjust the Discussion and Abstract if needed.
            Answer: Thank you for this comment. We agree with the reviewer and modified the texts as requested in the result and discussion section.

14) Similarly, on lines 320-322 the text refers to non-significant data and can be deleted or should be rewritten.     
 Answer: We changed the sentence accordingly to “[...] concentrations tended to be higher [...]” to match the level of association.

Reviewer 2 Report

The paper is well written and represents interesting translational research.

Minor corrections will improve its' clearness for the reader.

Minor corrections:

Authors: Sophia Kreft1,5 – should use affiliation numbers 1 and 2; affiliations should be numbered and used subsequently, not 5 before 2.

Line 108: A previous study suggested absolute ECP serum levels as a novel prognostic marker by analyzing a heterogeneous patient cohort [45]. – this sentence should be placed within the literature review part of the introduction; do not mix the aims of the study and literature review.

Line 114: please clearly describe the inclusion and exclusion criteria of patients recruited in the study i.e. line of treatment, use of adjuvant therapy.

Line 163: flow cytometry – please provide the model of FACS equipment.

Line: 165: (eBioscience) – ad city and country.

Line 172: anti-CD16-FITC or -PB, anti-CD66b-APC, anti-CD193-PE or -APC-Cy7 – give the name of the provider (company).

Line 210: please describe the method of co-culture (inserts? or mixed culture?); name (company) inserts used for transwell experiments.

Line: 212: Greiner Bio-One – give the name of the provider (company)

Introduction: Please cite data:

Changes in blood eosinophilia during anti-PD1 therapy as a predictor of long term disease control in metastatic melanoma. J Clin Oncol 33, 2015 (suppl; abstr 9069)

Pretreatment levels of absolute and relative eosinophil count to improve overall survival (OS) in patients with metastatic melanoma under treatment with ipilimumab, an anti CTLA-4 antibody. - J Clin Oncol 31, 2013 (suppl; abstr 9024)

Prognostic markers for progression-free survival (PFS) to anti PD-1 therapies in metastatic melanoma. J Clin Oncol 36, 2018 (suppl; abstr e21527)

Methods: Please check the methods section for consistency in listing the names of companies with city and country of reagents and equipment providers.

Discussion: In the discussion section please analyze the impact of samples collection timing on the results obtained: “White blood cell count and serum lactate dehydrogenase (LDH) were 131 assessed 0-63 days prior to collection of pre-treatment ECP samples and 0-63 days prior 132 to collection of on-treatment serum ECP samples”. “For samples from patients receiving dual targeted therapy the median of 126 days between first sample (pre-treatment) and second (on-treatment) sample was 98 days 127 (range 58-178 days). On-treatment serum samples was obtained close to the first response 128 assessment. For samples from patients receiving immunotherapy, the median time span 129 was 175 days (range 52 - 269 days).” Please describe expected time-dependent changes of eosinophile count during treatment, including its relation to drug infusions for immunotherapy.  – Please discuss i.e. Changes in blood eosinophilia during anti-PD1 threapy as a predictor of long term disease control in metastatic melanoma.  J Clin Oncol 33, 2015.  Gaba, Victoria, Pineda, et al.

Please discuss why non-adherent and adherent models were used. Which model better mimics in vivo melanoma tumor microenvironment and interactions between melanoma cells and eosinophils? Discuss i.e. “A new dawn for eosinophils in the tumor microenvironment” “Emerging Roles for Eosinophils in the Tumor Microenvironment”.

Please discuss how the eosinophil count reported in this study is related to AEs. Discuss i.e.: https://www.mdpi.com/2673-5601/1/3/17

Author Response

Reviewer 2:

Minor comments:

Authors: Sophia Kreft1,5 – should use affiliation numbers 1 and 2; affiliations should be numbered and used subsequently, not 5 before 2.        
            Answer: Thank you for this comment. We followed the advice of the reviewer and changed  the affiliation numbers.

Line 108: A previous study suggested absolute ECP serum levels as a novel prognostic marker by analyzing a heterogeneous patient cohort [45]. – this sentence should be placed within the literature review part of the introduction; do not mix the aims of the study and literature review.
            Answer: Thank you for pointing it out. We followed the advice of the reviewer and placed the sentence into the literature review. Additionally, we adjusted the sentence in the aim of the study.

Line 114: please clearly describe the inclusion and exclusion criteria of patients recruited in the study i.e. line of treatment, use of adjuvant therapy.          
            Answer: We apologize for the lack of clarity. 2.1 in the material and methods section was adjusted, hoping to explain the inclusion criteria for our study more clearly.

Line 163: flow cytometry – please provide the model of FACS equipment.     
            Answer: We added the respective information to the material and methods section. Flow cytometric measurements were performed using the CantoTM II FACS device from BD.

Line: 165: (eBioscience) – ad city and country.       
            Answer: We followed the request of the reviewer and added the city and country for the respective antibody.

Line 172: anti-CD16-FITC or -PB, anti-CD66b-APC, anti-CD193-PE or -APC-Cy7 – give the name of the provider (company).      
            Answer: As requested, we added the missing information about the provider, city and country for the respective antibodies.

Line 210: please describe the method of co-culture (inserts? or mixed culture?); name (company) inserts used for transwell experiments.           
            Answer: The sentence was changed and information about the insert used for Transwell experiments was added hoping to clarify the kind of co-cultures that were performed.

Line: 212: Greiner Bio-One – give the name of the provider (company)           
            Answer: The missing information about the provider was added to the text. The same specification was added in section 2.9. Colony Formation Assay. “After 20 minutes of incubation, triplicates were transferred to a 96-well plate (Greiner Bio-One, Cellstar®, Frickenhausen, Germany).”

Introduction: Please cite data:

Changes in blood eosinophilia during anti-PD1 therapy as a predictor of long term disease control in metastatic melanoma. J Clin Oncol 33, 2015 (suppl; abstr 9069)       
            Answer: Reference was added and cited in the Introduction section as requested by the reviewer. Citation can be found in the following sentence: “An early increase of eosinophil counts during anti-PD-1/anti-PD-L1 treatment was associated with improved outcome.“

Pretreatment levels of absolute and relative eosinophil count to improve overall survival (OS) in patients with metastatic melanoma under treatment with ipilimumab, an anti CTLA-4 antibody. - J Clin Oncol 31, 2013 (suppl; abstr 9024)  
            Answer: Reference was added and cited in the Introduction section as requested by the reviewer. Citation can be found in the following sentence: “In melanoma, the eosinophil count has been shown to positively correlate with survival and better response to immune checkpoint inhibition (ICI) or IL-2.“

Prognostic markers for progression-free survival (PFS) to anti PD-1 therapies in metastatic melanoma. J Clin Oncol 36, 2018 (suppl; abstr e21527)          
            Answer: Reference was added and cited in the Introduction section as requested. Citation can be found in the following sentence: “In melanoma, the eosinophil count has been shown to positively correlate with survival and better response to immune checkpoint inhibition (ICI) or IL-2.“ 

Methods: Please check the methods section for consistency in listing the names of companies with city and country of reagents and equipment providers.
            Answer: We apologize for the inconsistency. We added the information of the companies and their location as requested.

Discussion: In the discussion section please analyze the impact of samples collection timing on the results obtained: “White blood cell count and serum lactate dehydrogenase (LDH) were 131 assessed 0-63 days prior to collection of pre-treatment ECP samples and 0-63 days prior 132 to collection of on-treatment serum ECP samples”. “For samples from patients receiving dual targeted therapy the median of 126 days between first sample (pre-treatment) and second (on-treatment) sample was 98 days 127 (range 58-178 days). On-treatment serum samples was obtained close to the first response 128 assessment. For samples from patients receiving immunotherapy, the median time span 129 was 175 days (range 52 - 269 days).” Please describe expected time-dependent changes of eosinophile count during treatment, including its relation to drug infusions for immunotherapy.  – Please discuss i.e. Changes in blood eosinophilia during anti-PD1 threapy as a predictor of long term disease control in metastatic melanoma.  J Clin Oncol 33, 2015.  Gaba, Victoria, Pineda, et al.   
            Answer: As suggested, a sentence was added to the discussion section to refer to this topic. The proposed reference was cited and added.

Please discuss why non-adherent and adherent models were used. Which model better mimics in vivo melanoma tumor microenvironment and interactions between melanoma cells and eosinophils? Discuss i.e. “A new dawn for eosinophils in the tumor microenvironment” “Emergig Roles for Eosinophils in the Tumor Microenvironment”.  
            Answer: This is now addressed in the discussion section and the suggested references were added.

Please discuss how the eosinophil count reported in this study is related to AEs. Discuss i.e.: https://www.mdpi.com/2673-5601/1/3/17
            Answer: The link between eosinophil count and irAEs is now addressed in the discussion section. The corresponding reference was added.

Round 2

Reviewer 1 Report

My previous comments have been addressed. One minor item that arose with the new info is that it may seem surprising that the negative selection cocktail contains anti-CD123 (IL3 receptor alpha). Presumably, that is in order to remove basophils. However, eosinophils do express CD123 (see, e.g., Kelly EA et al. 2017, Am. J. Respir. Crit. Care Med. 196:1385-95), although at a low level, presumably too low level for the majority of eosinophils to be removed. But in theory, the inclusion of anti-CD123 may risk removing a subset of eosinophils with the highest CD123 expression and such a subset may differ among conditions. It would be interesting to have the authors comment on this.

Author Response

Thank you for this relevant comment.    When purifying eosinophils by negative selection using the Miltenyi kit, we also collected the population that contains all cells positive for the antibody-cocktail (positive fraction). When performing purity control stainings of the positive fraction, gating for CD66b positive granulocytes showed that >99% of these are CD16+ and only a tiny portion is CD16-. As the number of “left-over” eosinophils in the “CD123-positive fraction is tiny, we think that the loss of potential eosinophils is neglectable.